# DSI-Bench: A Benchmark for Dynamic Spatial Intelligence

**Fixed Observer and Object:**

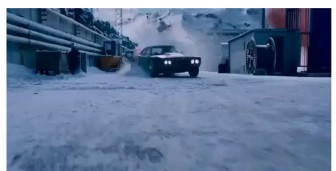

**Static Spatial Relation:**
Where is the car located in the 3D scene?
Where is the observer located in the 3D scene?
What's the absolute distance between the car and the observer?
Where is the observer positioned relative to the car?
……

**Observer and Object both in Motion:**

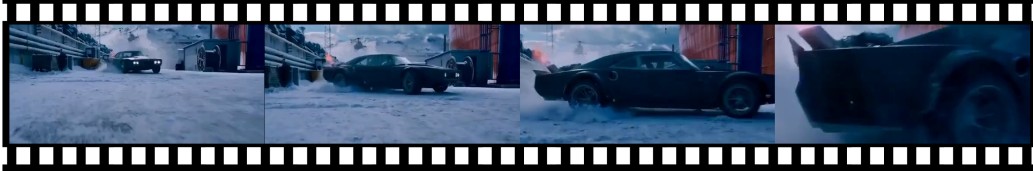

**Dynamic Object-Scene Relation:**
In this video, how does the car move in the 3D scene relative to its own starting orientation and position?
🧑 **Human: Moves forward and turns left.**
🤖 **VLM:**
**The car is moving forward and steering toward the right side of the frame, so it's moving forward and turning right.**

**Dynamic Observer-Scene Relation:**
How does the observer move in the 3D scene relative to its own starting orientation and position?
🧑 **Human: Move forward.**
🤖 **VLM:**
**The car is moving from the left side to the right, this relative motion indicates that the observer is moving in the opposite direction, specifically, moving leftward.**

**Dynamic Object-Scene Relation:**
How does the observer's position change relative to the car?
🧑 **Human:**
**Move from in front of the car to the car's right side.**
🤖 **VLM:**
**Initially, the observer is in front of the car; as it moved forward, more of the left side appeared, showing the observer shifted to the car's left.**

Figure 1: overview

## ABSTRACT

Reasoning about dynamic spatial relationships is essential, as both observers and objects often move simultaneously. Although vision-language models (VLMs) and visual expertise models excel in 2D tasks and static scenarios, their ability to fully understand dynamic 3D scenarios remains limited. We introduce **D**ynamic **S**patial **I**ntelligence and propose DSI-Bench, a benchmark with nearly 1,000 dynamic videos and over 1,700 manually annotated questions covering nine decoupled motion patterns of observers and objects. Spatially and temporally symmetric designs reduce biases and enable systematic evaluation of models' reasoning about self-motion and object motion. Our evaluation of 14 VLMs and expert models reveals key limitations: models often conflate observer and object motion, exhibit semantic biases, and fail to accurately infer relative relationships in dynamic scenarios. Our DSI-Bench provides valuable findings and insights about the future development of general and expertise models with dynamic spatial intelligence.

# 1 INTRODUCTION

When early humans chased prey across open landscapes, they instinctively adjusted their running paths in response to the animals' movements, helping them close the distance. Likewise, modern drivers adapt their direction and speed in response to the movements of other vehicles and pedestrians on the road. We live in a world where both the observer and the objects being observed are constantly in motion. For humans, understanding how our own position shifts and how other objects move in 3D space simultaneously is an intuitive ability rooted in basic spatial intelligence.

In recent years, significant progress has been made in visual perception and understanding. Vision-language models (VLMs), which deeply integrate visual and linguistic modalities, have demonstrated remarkable cross-modal semantic alignment and reasoning abilities, achieving impressive results in open-domain perception and dialogue. State-of-the-art VLMs OpenAI (2025a;b); Comanici et al. (2025); Wang et al. (2025b); Bai et al. (2025); ByteDance (2025) have achieved leading performance on 2D computer vision tasks such as detection and visual question answering, while also demonstrating a certain degree of understanding of temporal actions and static spatial relations. With the development of visual foundation models such as DINOv2 Oquab et al. (2023), an increasing number of studies introduce expertise models that exhibit strong zero-shot capabilities in spatial perception. These models demonstrate superior performance and robustness in accomplishing specific tasks and perceiving task-relevant information. For example, Dust3R Wang et al. (2024a), VGGT Wang et al. (2025a) enable robust estimation of camera poses and 3D motion trajectories, while the CoTracker Karaev et al. (2024) and SpatialTracker Xiao et al. (2025) families support pixel-level motion understanding, enabling highly accurate tracking of object and region trajectories in videos. These advances lay a critical foundation for building multimodal systems with spatial awareness and dynamic understanding.

However, current methods are still largely limited to scenarios where either the observer or the objects remain static. In more realistic and practically relevant dynamic environments, where both the observer and the objects are in motion, their performance and adaptability have not yet been systematically explored. To fill this gap, we introduce the problem of **D**ynamic **S**patial **I**ntelligence, and conduct a thorough, decoupled analysis of the ability of state-of-the-art methods to understand different forms of motion.

Having dynamic spatial intelligence entails the ability to decouple reasoning about both the agent's own motion and that of other objects within a scene. This requires models to exhibit a range of sophisticated capabilities, including temporal reasoning, capturing spatially consistent elements during dynamic changes, and understanding spatial relationships, which further encompasses spatiotemporal understanding and reasoning.

To this end, we propose DSI-bench, a VQA benchmark dedicated to Dynamic Spatial Intelligence. DSI-Bench comprises nearly 1000 dynamic scene videos which collected, cleaned, and clipped from diverse sources, covering 5 decoupled motion patterns of observers and objects. Focusing on the relationships among the observer, objects, and the scene, we designed 6 question types with over 1,700 VQA questions, all of which are manually annotated and verified. To mitigate biases in 3D space—such as left/right biases and correlations between object orientation and motion, we construct spatially and temporally symmetric versions of the same questions.

Our evaluation of 14 widely used VLMs and expertise models on DSI-Bench yields several key findings. By analyzing the models' self-explanations and visualization outputs, we observe that: 1) VLMs tend to conflate observer and the observed object's motion rather than treating them as distinct. 2) Semantic biases associated with the observed object can distort visual perception, leading to hallucinations. 3) Classical 3D constraints fail to consistently characterize relative pose relationships in continuous dynamic scene videos.

Our contribution can be summarized as:

- We introduce a more realistic and practically valuable Dynamic Spatial Intelligence task and propose DSI-Bench, a benchmark specifically designed for the systematic evaluation of dynamic spatial reasoning.

- We evaluate widely used general visual-language models and expertise visual foundation models on DSI-Bench and, through spatially and temporally symmetric sample designs, analyze their biases and hallucination tendencies in dynamic scenarios.

- Finally, by examining failure cases of state-of-the-art spatial foundation models, we highlight the limitations of classical 3D constraints in dynamic spatial perception.

## 2 RELATED WORKS

### 2.1 LARGE VISION-LANGUAGE MODEL

Beyond single-frame understanding, VLMs OpenAI (2025a;b); ByteDance (2025); Comanici et al. (2025) have advanced to capture both temporal information and static spatial relations in videos, achieving strong performance across tasks such as detection, open dialogue, and multimodal reasoning. For instance, the Qwen2.5-VL Bai et al. (2025) series demonstrates strong performance in long video understanding by incorporating absolute timestamp encoding. Its innovative visual encoder further allows dynamic adjustment of frame rates and video resolution, enhancing applicability across diverse video scenarios. Similarly, the InternVL-3.5 Wang et al. (2025b) series employs a visual resolution router that dynamically focuses on fine-grained video details, significantly improving video reasoning efficiency while maintaining overall performance.

### 2.2 BENCHMARKS FOR SPATIAL INTELLIGENCE

To evaluate and investigate the spatial intelligence of VLMs, a series of benchmarks have been proposed. VSI-Bench Yang et al. (2025a) systematically measures model performance across 8 categories of spatial reasoning tasks and examines how cognitive maps may facilitate VLM inference. MMSI-Bench Yang et al. (2025b) design problems that cannot be solved from a single viewpoint, highlighting the importance of multi-image dependency in spatial reasoning. VLM4D Zhou et al. (2025) extends this line of work by treating video as a four-dimensional modality, thereby exploring the models' capacity to capture spatiotemporal correlations.

Although these benchmarks provide a foundation for assessing spatial intelligence, they overlook issues of multimodal hallucination and bias in VLMs. Recently, 3DSR-Bench Ma et al. (2024) addressed this issue by introducing the Evalflip strategy and designing queries from uncommon perspectives to analyze hallucination in more diverse settings. Similarly, Ori-Bench Wang et al. (2024c) revealed that VLMs are prone to orientation hallucinations when misled by the semantic priors of observer-centric orientations.

Nevertheless, most existing evaluations focus on static scenes or observers, limiting relevance to real-world dynamics. To address this, we propose DSI-Bench, which treats static cases as special forms of dynamic scenarios. DSI-Bench encompasses a rich video dataset that disentangles 5 motion types for both observers and the observed object, thereby covering a broader and more realistic range of applications. Furthermore, we generate diverse and complementary samples through spatial–temporal symmetry-based augmentation and analyze model bias and hallucination via multiview statistical evaluation.

### 2.3 3D VISUAL SPATIAL EXPERTISE MODELS

The combination of classical 3D constraints with modern 2D foundation models has markedly advanced the performance and efficiency of spatial expertise models. For example, VGGT Wang et al. (2025a) leverages DINOv2 Oquab et al. (2023) features and combines depth estimation, keypoint tracking, and pose estimation, which together enable it to reconstruct complete 3D scenes efficiently from a sequence of images. Building on this, SpatialTrackerV2 performs joint reasoning across decoupled spatial factors such as scene geometry, camera poses, and object motion, thereby enhancing its ability to track dynamic object trajectories. However, classical geometric constraints Schönberger & Frahm (2016); Hartley & Zisserman (2003); Schönberger et al. (2016); Furukawa et al. (2015) are primarily designed for static scenes. When both the observer and the environment are in motion, these constraints introduce instability into keypoint tracking and distance estimation.

## 3 DSI-BENCH

In this section, we introduce DSI-Bench, designed to evaluate models' ability to perceive observer and object motion in realistic dynamic videos. Section 3.1 proposes the concept of Dynamic Spatial Intelligence along with its associated sub-tasks. Section 3.2 provides a detailed account of the benchmark construction, covering data collection and standardization, spatio-temporal flip augmentation, and the design of QA tasks.

### 3.1 OVERVIEW

In dynamic scenarios, both the observer and the observed object may be in motion. Under such conditions, the spatial relationships commonly considered in static scenes—such as the positions of the observer and objects in 3D space, as well as their relative distances and orientations—are further extended into temporal variations, including changes in position, distance, and orientation. As illustrated in Figure. 1, we refer to the perception and reasoning of these spatio-temporal relationships collectively as **D**ynamic **S**patial **I**ntelligence.

We categorize tasks based on three fundamental 3D entities: the observer, the observed object, and the scene. This yields three types of tasks:

1. **Object–Scene tasks**: examine spatial relationships between objects and the scene, distinguishing between cases where the observer is moving versus stationary.

2. **Observer–Scene tasks**: evaluate the ability to track changes in the observer's 3D pose under both dynamic and static conditions.

3. **Observer–Object tasks**: focus on the relative relationship between the observer and the observed object, with typical questions involving the estimation of distance or orientation changes.

Building on this task taxonomy, we constructed over 1,700 VQA pairs that comprehensively cover the above categories of dynamic spatial relationships. Detailed statistics of tasks and examples are reported in Figure 2.

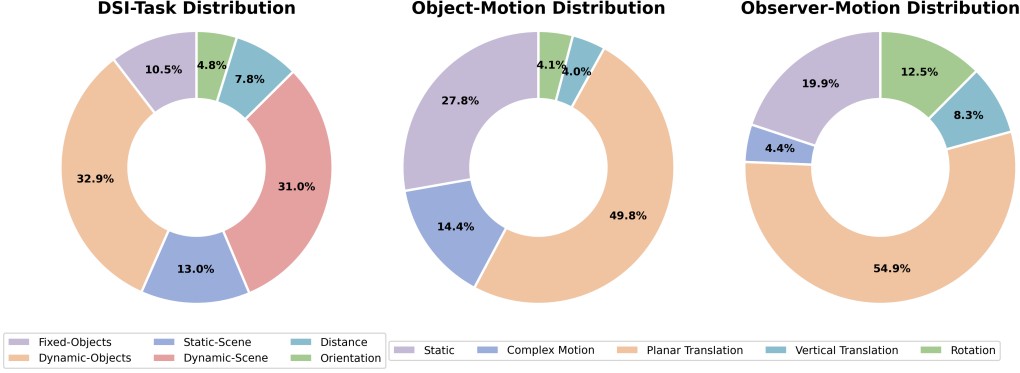

Figure 2: Left: Task distribution in DSI-Bench; Middle: Observer Motion distribution in DSI-Bench; Right: Observed Motion distribution in DSI-Bench

### 3.2 BENCHMARK CONSTRUCTION PROCESS

The details of the DSI-Bench data construction pipeline are shown in Figure 3.

**Data Collection and Standardization** DSI-Bench comprises over 1,700 question-answer pairs derived from 943 videos, sampled from the camera motion dataset (CameraBench Lin et al. (2025)), the object motion dataset (Kinetics-700 Smaira et al. (2020)), and the synthetic motion-control dataset (SynFMC Shuai et al. (2025)). To further increase the diversity of observer and object motion patterns, we also supplemented the dataset with videos from LLaVA-178K Zhang et al. (2024) and additional online sources. This diverse collection ensures that DSI-Bench captures a wide range of

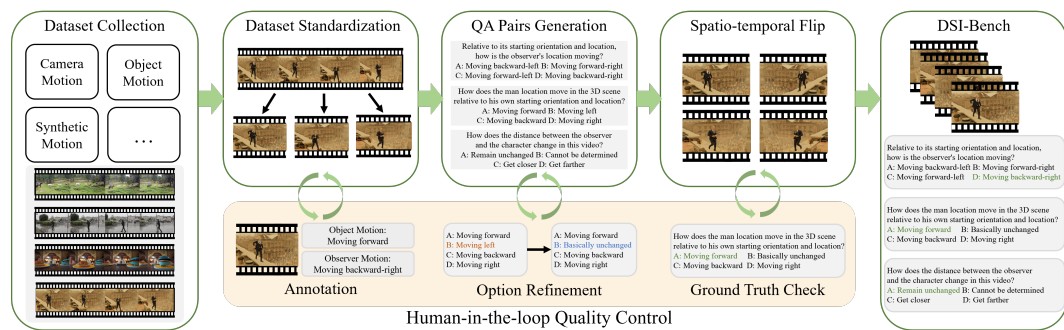

Figure 3: Illustration of the DSI-Bench construction pipeline: videos are sampled from diverse motion datasets, QA tasks and options are template-based genrated and manually refined, and videos are augmented with spatio-temporal flipping to mitigate data bias.

dynamic relationships. Specifically, both the observer and the observed objects exhibit three types of motion: translation (e.g., forward or upward movement), rotation (clockwise or counterclockwise), and a combination of the two (e.g., forward-left or forward-right turns). We further applied spatio-temporal flip augmentation to balance the distribution of these motion types. See the middle and right panels of Figure 2 for detailed motion statistics of DSI-Bench.

For preprocessing, we used PySceneDetectCastellano to segment videos into scenes and applied SpatialTrackerV2 to filter out clips with irregular or jittery observer motion. The human experts then conducted the final selection and determined the starting and end points of each video. All videos are standardized to a resolution of 480p, and overly short clips were slowed down to a duration of 3 seconds.

**Question-Answer Generation.** Building on the standardized video data, we manually annotated the motion patterns of both the observer and the observed objects within the 3D scene. Using these annotations, we then applied a template-based approach to construct Cam-Scene and Obj-Scene VQA pairs. For a subset of videos, we additionally annotated the relative distance changes to generate relative-distance VQA pairs. All observed objects were further annotated with orientation information, which enabled the construction of relative-orientation VQA pairs.

To avoid ambiguity caused by shifting reference points in dynamic scenes, we followed the conventions of prior work Wang et al. (2025a); Xiao et al. (2025), and fixed the 3D reference point to the initial pose of either the observer or the observed object in each video. Finally, all VQA pairs were reviewed, filtered, and refined by human experts to ensure clarity and eliminate ambiguity.

**Spatio-Temporal Flip Augmentation** To mitigate potential biases in 3D motion patterns and to assess model robustness, we introduced spatio-temporal augmentations inspired by 3DSR-Bench Ma et al. (2024) and Ori-Bench Wang et al. (2024b). Each video is horizontally flipped first, and both the standard and flipped versions are further reversed in time, producing four variants in total: standard, horizontal flip, reverse, and reverse + horizontal flip.

For the corresponding VQA pairs, the questions remained unchanged, while the answer options are symmetrically adjusted using a rule-based method (e.g., "moving forward" → "moving backward" after reversal; "rotating clockwise" → "rotating counterclockwise" after flipping). This ensured that the ground truth labels remained consistent. However, since video reversal shifts the reference frame from the first to the last frame of the standard video, some samples could not be handled by rule-based substitution alone. These cases are manually inspected and corrected by human experts.

## 4 EVALUATION ON DSI-BENCH

### 4.1 EVALUATION SETUP

**Benchmark models.** We conduct a comprehensive evaluation of a wide range of VLMs, encompassing both proprietary and open-source families, diverse model scales, and recent architectural advances. For proprietary models, our benchmark includes Nova-pro-v1 Amazon (2025), GPT-

4o OpenAI (2025a), GPT-5 OpenAI (2025b), Gemini-2.5-Pro Comanici et al. (2025) , Seed-1.6, and Seed-1.6-vision ByteDance (2025). For open-source models, we assess state-of-the-art models Qwen2.5-VL Bai et al. (2025) series and InternVL-3.5 Wang et al. (2025b)series.

For the 3D expertise models, we selected two representative approaches, VGGT Wang et al. (2025a) and SpatialTrackerV2 Xiao et al. (2025), for evaluation on DSI-Bench.

**VLM Evaluation Details.** We measure each VLM's accuracy by directly comparing the model's selected answer with the ground truth, without employing any additional external models or annotations for performance evaluation. For consistency, all models are evaluated with a temperature of zero, a maximum output length of 2,048 tokens, and a video sampling rate of 5 fps.

To investigate the impact of reasoning on dynamic spatial intelligence and to analyze potential performance bottlenecks of VLMs, we adopt the MindCube Yin et al. (2025) protocol and evaluate all models under two distinct settings: Direct Answering (RAWQA) and Free-Form Reasoning (FFR). In RAWQA, models are required to output only the final answer without any intermediate content, whereas in FFR, models must articulate their reasoning process before answering. The corresponding system prompts are provided in the Appendix.

| Models | Object-Scene | | Observer-Scene | | Observer-Object | | Overall |
|---|---|---|---|---|---|---|---|
| | Fixed-Obs. | Dyn-Obs. | Static-Sce. | Dyn-Sce. | Distance | Orientation | |
| Random | 25.00% | 25.00% | 25.00% | 25.00% | 25.00% | 25.00% | 25.00% |
| **Sample-wise Evaluation** | | | | | | | |
| Gemini-2.5-Pro | 45.54% | 44.76% | 54.89% | 40.39% | 69.75% | 47.94% | 46.90% |
| Nova-Pro-V1 | 38.92% | 37.33% | 34.57% | 28.64% | 46.01% | 15.29% | 34.06% |
| Qwen2.5-VL-32B | 37.16% | 37.54% | 29.46% | 34.06% | 58.15% | 32.35% | 36.73% |
| Qwen2.5-VL-72B | 41.35% | 41.92% | 33.26% | 34.56% | 58.15% | 39.71% | 39.61% |
| Seed-1.6 | 45.27% | 45.15% | 35.00% | 35.52% | 54.17% | 41.47% | 41.38% |
| Seed-1.6-Vision | 45.54% | 42.87% | 50.76% | 39.21% | 79.71% | 38.53% | 45.70% |
| GPT-4o | 42.43% | 39.65% | 37.61% | 28.78% | 55.98% | 32.35% | 37.23% |
| GPT-5 | 43.37% | 36.73% | 39.13% | 34.61% | 73.55% | 40.59% | 40.14% |
| InternVL3.5-8B | 39.73% | 37.97% | 26.20% | 32.60% | 62.14% | 29.12% | 36.41% |
| InternVL3.5-38B | 42.02% | 37.63% | 34.57% | 34.70% | 67.39% | 37.65% | 39.10% |
| InternVL3.5-30BA3B | 42.70% | 39.13% | 35.43% | 32.10% | 60.14% | 34.12% | 38.24% |
| InternVL3.5-241BA30B | 44.06% | 43.51% | 37.28% | 33.88% | 61.78% | 30.88% | 40.59% |
| VGGT | – | – | 35.55% | 22.50% | – | – | – |
| SpatialTrackerV2 | 25.89% | 27.84% | 42.72% | 40.12% | 26.42% | – | 34.97% |
| **Group-wise Evaluation** | | | | | | | |
| Random | 0.05% | 0.05% | 0.05% | 0.05% | 0.05% | 0.05% | 0.05% |
| Gemini-2.5-Pro | 21.62% | 18.21% | 42.61% | 21.86% | 64.49% | 31.76% | 27.13% |
| Nova-Pro-V1 | 13.52% | 10.82% | 18.26% | 8.56% | 38.41% | 5.88% | 13.29% |
| Qwen2.5-vl-32B | 10.81% | 10.48% | 14.78% | 16.39% | 50.72% | 17.65% | 16.40% |
| Qwen2.5-vl-72B | 12.44% | 10.31% | 15.65% | 10.93% | 46.38% | 35.29% | 15.43% |
| Seed-1.6 | 16.76% | 13.40% | 15.22% | 10.93% | 42.75% | 25.88% | 16.11% |
| Seed-1.6-Vision | 18.38% | 13.23% | 39.13% | 21.49% | 74.64% | 30.59% | 25.33% |
| GPT-4o | 14.06% | 10.31% | 24.78% | 9.29% | 45.65% | 20.00% | 15.49% |
| GPT-5 | 17.84% | 13.23% | 26.96% | 17.49% | 70.29% | 25.88% | 21.88% |
| InternVL3.5-8B | 19.46% | 12.03% | 13.04% | 19.13% | 50.72% | 10.59% | 18.08% |
| InternVL3.5-38B | 22.16% | 12.03% | 18.26% | 11.84% | 58.70% | 28.24% | 18.25% |
| InternVL3.5-30BA3B | 22.70% | 12.03% | 22.61% | 18.21% | 43.48% | 23.53% | 19.44% |
| InternVL3.5-241BA30B | 16.76% | 14.60% | 19.57% | 11.48% | 49.28% | 18.82% | 17.41% |
| VGGT | – | – | 31.30% | 16.58% | – | – | – |
| SpatialTrackerV2 | 17.39% | 19.35% | 40.00% | 37.89% | 11.49% | – | 28.34% |

Table 1: Evaluation results for 14 models. Sample-wise accuracy treats all augmented videos as independent; group-wise reports the fraction of video groups with at least 3 correct predictions among augmented variants of the same original video.

**3D Expertise Model Evaluation Details.** We manually calibrated the orientations of the observed objects in the videos and accordingly removed benchmark samples that specifically tested orientation reasoning. We also refined the masks predicted by Segment Anything to align keypoints with the corresponding observed objects. The predicted trajectories of both the observer and the observed objects are then extracted and mapped to the corresponding answer choices through a rule-based procedure.

**Spatio-Temporal Flip Evaluation** In Section 3.2, we introduce the method of constructing augmented samples through spatio-temporal flipping. In testing, we employ two evaluation strategies, Sample-wise and Group-wise, to assess model performance and robustness. In the former, each spatio-temporal flip video is treated as an independent sample and evaluated separately; in the latter, the four flip variations are grouped as one instance, which is counted correct only if at least three answers are correct. The experimental results under the two strategies are presented in the upper and lower halves of Table 3, respectively.

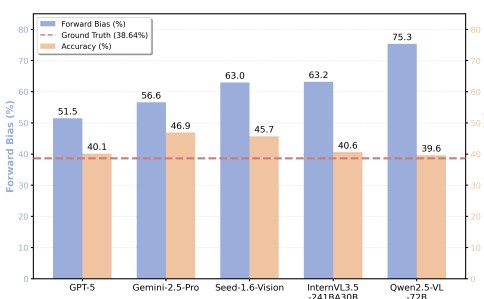

Figure 4: Proportion of models selected with the "forward" option and ground-truth annotations containing "forward".

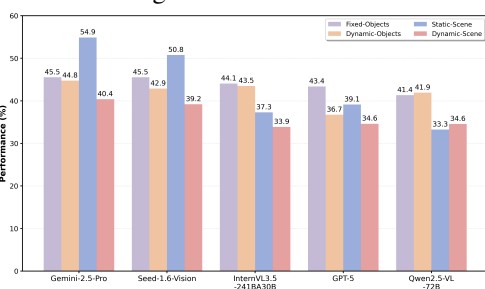

Figure 5: Performance gap of VLMs between static and dynamic conditions.

Table 2: Impact of free-form reasoning on DSI-Bench.

| Models | Overall Accuracy | |
|---|---|---|
| | RAWQA | FFR ($\Delta$) |
| Gemini-2.5-pro | 46.9% | 46.0% (-0.9%) |
| Nova-Pro-V1 | 34.1% | 36.9% (+2.8%) |
| Qwen2.5-VL-32B | 36.7% | 39.5% (+2.8%) |
| Qwen2.5-VL-72B | 39.6% | 39.8% (+0.2%) |
| Seed-1.6 | 41.4% | 41.8% (+0.4%) |
| Seed-1.6-vision | 45.7% | 45.7% (-0.0%) |
| GPT-4o | 37.2% | 38.4% (+1.1%) |
| GPT-5 | 40.1% | 40.5% (+0.4%) |
| InternVL3.5-8B | 36.4% | 35.7% (-0.7%) |
| InternVL3.5-38B | 39.1% | 38.6% (-0.5%) |
| InternVL3.5-30BA3B | 38.2% | 38.2% (-0.1%) |
| InternVL3.5-241BA30B | 40.6% | 38.5% (-2.0%) |

## 4.2 MAIN RESULTS

Table 3 presents the main results of the evaluation in DSI-Bench, covering both proprietary and open source VLMs, as well as 3D expertise models. We summarize the key findings below:

**Current models struggle more with dynamic than with static Tasks.** We present the performance difference of the model under dynamic and static conditions in Figure 5, and observe that dynamic scenarios consistently present greater challenges than static ones. When either the observer or the observed object is in motion, most models experience a notable decline in accuracy on tasks that are otherwise equivalent. This performance degradation is especially evident in models with relatively advanced spatial reasoning capabilities. For example, in Seed-1.6-vision, the accuracy in understanding object dynamics decreases by 2.67% when the observer is moving compared with a static observer, while the accuracy in perceiving observer motion drops by as much as 11.55% under dynamic conditions relative to static ones.

Group-wise evaluation reinforces this observation: models show reduced robustness in dynamic settings, and the performance gap between static and dynamic samples becomes even more pronounced.

**Free-form reasoning yields only marginal and unstable benefits.** Table 2 summarizes the performance difference between the model's direct answers and its free-form reasoning. For most models, enabling free-form reasoning leads to only minor and inconsistent improvements. For instance, QwenVL2.5-72B achieves merely a 0.17% increase in overall accuracy under free-form reasoning mode. Some models even perform worse: Gemini-2.5-Pro and InternVL3.5-241B exhibit lower accuracy compared with directly producing answers. We attribute this to the fact that current VLMs primarily ground their reasoning on information extracted by the visual encoder. As a result, language-based reasoning alone cannot compensate for errors that originate from inaccurate visual perception.

Closer inspection of VLMs' reasoning process further reveals that VLMs often rely on common-sense knowledge within the language modality, which sometimes introduces additional biases and hallucinations into visual reasoning. A more detailed analysis of this phenomenon is provided in Section 5.1. Notably, in certain cases, models fail to terminate their reasoning process and instead generate incoherent text until reaching the maximum output token limit.

**VLMs exhibit limited robustness.** By comparing sample-wise and group-wise accuracy, we observe substantial performance drops across all task categories for VLMs. In contrast, 3D expertise models such as SpatialTrackerV2 show minimal degradation under group-wise evaluation. We hypothesize that VLMs fail to accurately perceive dynamic spatial information in videos; instead, they exhibit certain biases and hallucinations that impair their performance.

**Scaling up model size enhances accuracy but not robustness.** Larger models generally achieve higher overall performance, but this improvement does not translate into greater robustness. We observe that, within the same VLM architecture, increasing the number of parameters yields better results on DSI-Bench in sample-wise evaluation. For example, QwenVL2.5-72B surpasses its 32B counterpart by 2.8%, InternVL3.5-38B outperforms the 8B version by 2.69%, and InternVL3.5-241B-A28B exceeds InternVL3.5-30B-A3B by 2.35%. Expanding model size enables models to capture finer-grained details, thereby improving perceptual accuracy.

However, group-wise evaluation, which places greater emphasis on robustness, reveals the opposite trend: QwenVL2.5-72B, InternVL3.5-38B, and InternVL3.5-241B-A28B all underperform their smaller counterparts (32B, 8B, and 30B-A3B, respectively). This suggests that model size may not be the principal bottleneck for video-based spatial intelligence in current VLMs. While larger models enhance perceptual ability, they do not eliminate inherent biases in spatial perception and reasoning patterns.

## 5 ERROR ANALYSIS

### 5.1 BIAS AND HALLUCINATION ON VLM

To identify the performance bottlenecks of VLMs, we manually examined their response tendencies and reasoning processes, analyzing the underlying causes of errors. The following are some of our key observations:

**Forward Bias.** We selected all VQA pairs in DSI-Bench whose answer options contain the string "forward" and calculated the frequency with which VLMs chose these options. The results are presented in Figure 4. Our analysis reveals that the proportion of "forward" selections by the models far exceeds the true proportion of ground truths containing "forward," indicating a strong selection bias. The example of "fixed statues" shown in Figure 6 further illustrates how this "forward bias" interferes with visual perception, leading the models to generate multimodal hallucinations. We hypothesize that this bias is related to the imbalanced distribution of visual datasets, in which forward motion predominates over other movement patterns for animals, characters, vehicles, and similar entities.

**Undecoupled Motion Reasoning.** Confusion between the orientations and motions of the observer and the observed object constitutes another major source of error. We find that current VLMs are unable to independently infer the motion of the observer in 3D space and the motion of the observed object. Figure 8 illustrates two representative patterns of such entangled reasoning. In Figure

8a, when inferring the observer's motion, the model mistakenly substitutes the orientation and movement of the observed object for that of the observer, effectively assuming that the observer and the object are stationary relative to each other. Figure 8a shows an example in which this relative-motion error is erroneously generalized to the scene reference frame.

**Confuse rotation with translation** We identified a specific type of error in which VLMs misinterpret in-place rotations as translations when inferring the observer's motion. As illustrated in Figures 7, VLMs attempt to determine the direction of observer motion by reasoning about "which side of the scene enters the field of view." However, the models often fail to distinguish whether this visual change arises from a rotation around the observer or a translation through space. In these tasks, 3D expertise models that leverage classical geometric constraints for camera pose estimation achieve superior performance.

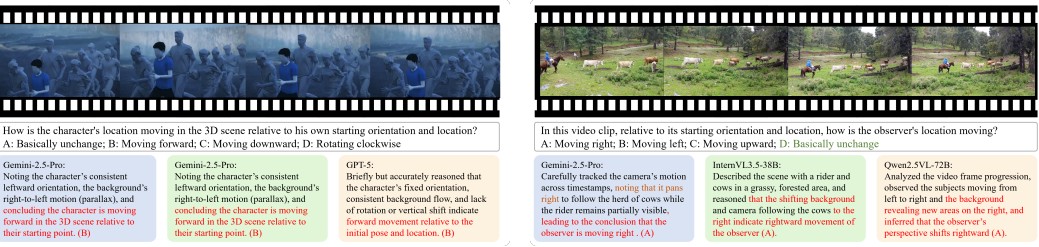

Figure 6: For fixed objects, VLMs hallucinated forward motion.

Figure 7: VLMs conflated translation and rotation, which are two distinct types of motion.

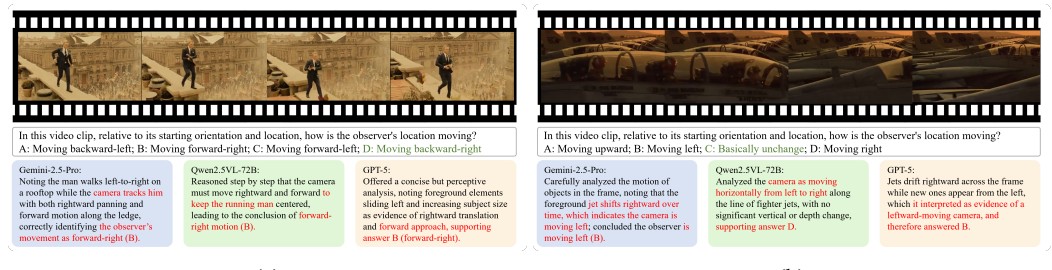

(a)                                                    (b)

Figure 8: The model fails to independently interpret the dynamics of the observer and the observed object, often relying on relative motion for indirect inference, which leads to hallucinations.

### 5.2 INSTABILITY ON 3D EXPERTISE MODEL

The results in Table 3 demonstrate that 3D expertise models exhibit robust camera pose estimation in dynamic scenarios, showing the smallest performance degradation under group-wise symmetric evaluation. However, their ability to track observed objects is less consistent. Experimental results indicate that in dynamic settings these models struggle to accurately estimate the relative distance between the observer and the observed object, which may in turn indirectly impair their accuracy in estimating object motion.

## 6 CONCLUSION

In this work, we study the dynamic spatial intelligence of models. We introduce a new benchmark, DSI-Bench, which comprises over 1,700 VQA pairs constructed from nearly 1,000 videos featuring diverse dynamic relationships. To mitigate motion-pattern biases inherent in 3D data, we adopt a spatio-temporal flipping strategy, and we assess model robustness using a group-wise evaluation protocol. We benchmark a range of open-source and proprietary VLMs, as well as 3D expert models, on DSI-Bench, and conduct statistical and error analyses to identify the sources of perceptual illusions and biases. We hope that DSI-Bench will serve as a valuable resource for advancing models' dynamic spatial perception and reasoning capabilities.

## 7 ETHICS STATEMENT

We have carefully reviewed the ICLR Code of Ethics and commit to strictly adhering to its guidelines in all aspects of our participation in the conference.

## 8 REPRODUCIBILITY STATEMENT

We ensure the reproducibility of our results. We will release our dataset and evaluation code after the review process.

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

# A APPENDIX

## A.1 FULL RESULTS OF VLM FREE-FORM REASONING

| Models | Object-Scene | | Observer-Scene | | Observer-Object | | Overall |
|---|---|---|---|---|---|---|---|
| | Fixed-Obs. | Dyn-Obs. | Static-Sce. | Dyn-Sce. | Distance | Orientation | |
| Random | 25.00% | 25.00% | 25.00% | 25.00% | 25.00% | 25.00% | 25.00% |
| **Sample-wise Evaluation** | | | | | | | |
| Gemini-2.5-Pro | 47.16% | 44.85% | 52.61% | 37.61% | 70.47% | 47.35% | 45.97% |
| Nova-Pro-V1 | 41.08% | 39.56% | 35.76% | 33.83% | 39.31% | 27.65% | 36.86% |
| Qwen2.5-VL-32B | 43.78% | 42.44% | 36.41% | 36.70% | 43.84% | 30.29% | 39.54% |
| Qwen2.5-VL-72B | 41.89% | 42.44% | 37.17% | 34.02% | 52.36% | 40.88% | 39.78% |
| Seed-1.6 | 43.51% | 45.58% | 37.50% | 36.89% | 48.55% | 43.82% | 41.76% |
| Seed-1.6-Vision | 42.16% | 40.25% | 53.26% | 40.12% | 83.33% | 44.71% | 45.68% |
| GPT-4o | 41.21% | 39.00% | 37.83% | 31.79% | 59.24% | 37.94% | 38.36% |
| GPT-5 | 43.51% | 36.81% | 38.26% | 35.61% | 75.36% | 40.88% | 40.53% |
| InternVL3.5-8B | 36.49% | 37.76% | 28.91% | 30.24% | 59.96% | 33.82% | 35.68% |
| InternVL3.5-38B | 38.52% | 40.12% | 35.98% | 33.79% | 59.06% | 33.82% | 38.62% |
| InternVL3.5-30BA3B | 40.81% | 39.69% | 35.43% | 33.33% | 55.07% | 33.24% | 38.17% |
| InternVL3.5-241BA30B | 41.22% | 42.65% | 35.87% | 32.88% | 51.45% | 27.35% | 38.54% |
| **Group-wise Evaluation** | | | | | | | |
| Random | 0.05% | 0.05% | 0.05% | 0.05% | 0.05% | 0.05% | 0.05% |
| Gemini-2.5-Pro | 21.62% | 17.35% | 40.87% | 20.40% | 63.77% | 31.76% | 26.11% |
| Nova-Pro-V1 | 21.08% | 16.15% | 25.65% | 17.49% | 28.99% | 18.82% | 19.45% |
| Qwen2.5-VL-32B | 21.62% | 15.64% | 19.57% | 19.49% | 30.43% | 12.94% | 19.00% |
| Qwen2.5-VL-72B | 14.59% | 15.12% | 21.74% | 11.11% | 40.58% | 29.41% | 17.35% |
| Seed-1.6 | 14.06% | 14.95% | 18.26% | 14.57% | 34.78% | 27.06% | 17.30% |
| Seed-1.6-Vision | 14.06% | 10.82% | 40.87% | 19.31% | 79.71% | 30.59% | 24.02% |
| GPT-4o | 19.46% | 14.26% | 26.09% | 14.39% | 52.17% | 22.35% | 19.73% |
| GPT-5 | 20.54% | 12.71% | 25.65% | 18.94% | 70.29% | 29.41% | 22.44% |
| InternVL3.5-8B | 14.60% | 10.48% | 9.13% | 10.93% | 47.83% | 18.82% | 14.19% |
| InternVL3.5-38B | 18.92% | 14.78% | 16.09% | 14.39% | 47.10% | 18.82% | 17.97% |
| InternVL3.5-30BA3B | 14.60% | 11.34% | 13.04% | 12.57% | 37.68% | 21.18% | 14.81% |
| InternVL3.5-241BA30B | 13.52% | 15.46% | 16.96% | 12.02% | 39.86% | 15.29% | 16.29% |

Table 3: Free-form Reasoning results for 12 VLMs. Sample-wise accuracy treats all augmented videos as independent; group-wise reports the fraction of video groups with at least 3 correct predictions among augmented variants of the same original video.

## A.2 SYSTEM PROMPT FOR RAWQA AND FFR

**RAWQA** ``You are a vision-language expert.
You are given a clip of video and your task is to answer a
question about the video.
You only need to provide *ONE* correct answer selecting from the
options listed below. For example, if you think the correct
answer is 'A' from 'A. Above B. Under C. Front D. Behind', your
response should **only** be '<answer>A</answer>'.
Please answer the question in this format strictly:
<answer>[replace your answer here, A, B, C, or D]</answer>''

**FFR** ``You are a vision-language expert.
You are given a clip of video and your task is to answer a
question about the video.
please provide your reasons step by step in details, then provide

```
*ONE* correct answer selecting from the options.
You only need to provide *ONE* correct answer selecting from the
options listed below.  For example, if you think the correct
answer is 'A' from 'A. Above B. Under C. Front D. Behind', your
response should only be '<answer>A</answer>'.
Please answer the question in this format strictly:
<think>[replace your reasoning here]</think>
<answer>[replace your answer here, A, B, C, or D]</answer>''
```

## A.3 ADDITIONAL VISUALIZATION EXAMPLES

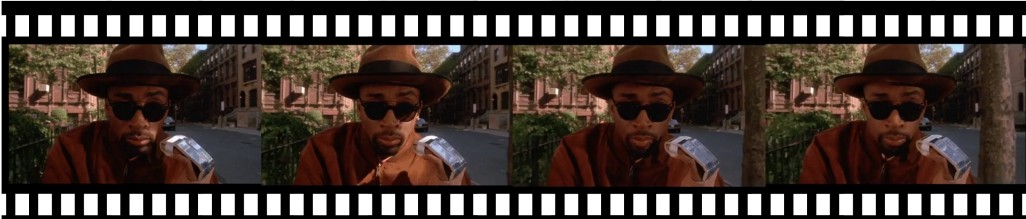

How is the character's location moving in the 3D scene relative to his/her/its starting orientation and location?
A: Moving right; B: Moving backward; C: Moving forward; D: Moving left

Gemini-2.5-Pro:
The background moves right-to-left while the character stays centered, indicating the camera and character move right relative to the scene. Since the character faces the camera, this direction corresponds to his left, so the answer is D.

Qwen2.5VL-72B:
Observes no movement in the character or background, states none of the options are accurate, yet still chooses B without justification.

GPT-5:
Notes the background shifts left with consistent scale, indicating lateral translation to the right; concludes the character (or camera) is moving right, selecting A.

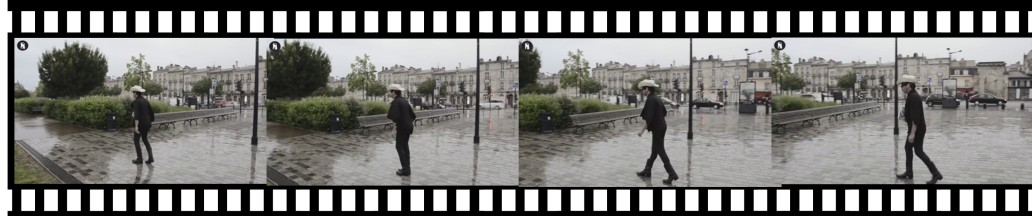

In this video clip, relative to its starting orientation and location, how is the observer's location moving?
A: Moving right; B: Rotating clockwise; C: Moving left; D: Rotating counterclockwise

Gemini-2.5-Pro:
The background shifts left as the camera tracks the person walking right, indicating the observer is moving right. This lateral translation matches option A, not rotation or leftward motion.

Qwen2.5VL-72B:
The observer moves from left to right without changing orientation, confirming rightward translation.

GPT-5:
Leftward scene shift with parallax (near objects move faster than distant ones) indicates the observer is translating right, not rotating.

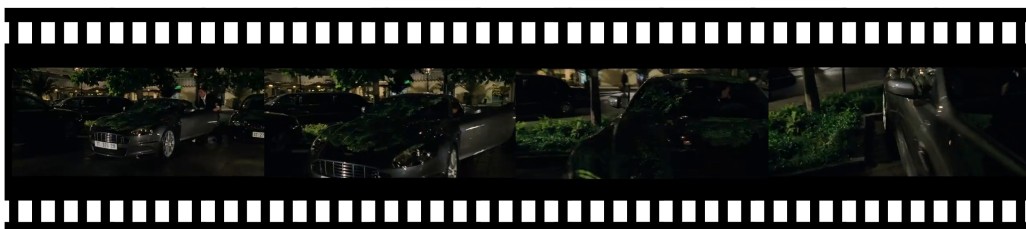

In this video clip, relative to its starting orientation and location, how is the observer's location moving?
A: Moving forward-right; B: Moving forward-left; C: Moving backward-left; D: Moving backward-right

Gemini-2.5-Pro:
Observes the camera moving leftward and backward relative to the car—front grille exits right, more background appears—concluding the motion is backward-left (C).

Qwen2.5VL-72B:
Assumes the observer tracks the car as it drives forward, placing the camera behind and to the right, thus selecting forward-right (A).

GPT-5:
Notes the car grows larger (suggesting forward motion) and the viewpoint shifts toward the rear-right side, interpreting this as forward-right movement (A).

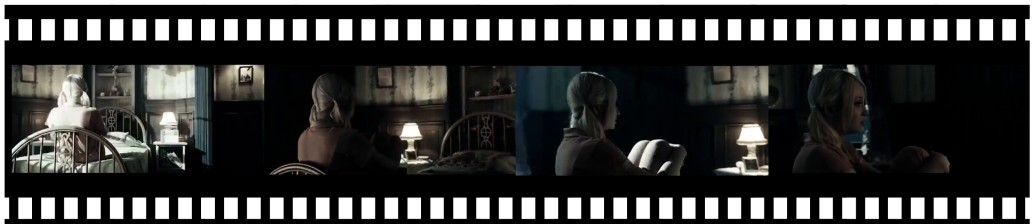

In this video clip, relative to its starting orientation and location, how is the observer's location moving?
A: Moving forward-right; B: Moving forward-left; C: Moving backward-left; D: Moving backward-right

Gemini-2.5-Pro:
The camera starts behind the character and arcs to her left, ending in a left-profile view; correctly identifies the motion as "from back to left" (C).

Qwen2.5VL-72B:
Notes the camera begins behind the character and gradually shifts leftward to capture her left side, supporting answer C.

GPT-5:
Tracks the movement from a centered rear view to the left side, using contextual cues like the lamp's position to confirm the observer ends up on the character's left (C).

## A.4 USE OF LLMS

We leverage LLMs to refine the expression and assist with the formatting of the paper. For summarizing experimental results, LLMs are employed to distill the reasoning process, facilitating the generation of more effective visualizations.

