# OpenReview forum: "DSI-Bench: A Benchmark for Dynamic Spatial Intelligence"
_ICLR.cc/2026/Conference — ICLR 2026 Conference Withdrawn Submission_

### Official Review · Reviewer_tXr9 · 2025-10-17

**Soundness:** 3
**Presentation:** 2
**Contribution:** 2
**Rating:** 4
**Confidence:** 4

**Summary:**

DSI-Bench is a benchmark for Dynamic Spatial Intelligence with ~1,000 videos and 1,700+ VQA pairs that disentangle observer and object motion via spatio-temporal symmetry. Evaluating 14 VLMs, it reveals consistent failures in dynamic settings—motion conflation, “forward” bias, weak robustness—and minimal gains from free-form reasoning.

**Strengths:**

- Dynamic Spatial Intelligence is a valuable research direction. This paper proposes a benchmark for Dynamic Spatial Intelligence with clear categorization and sufficient scale.
- It employs Spatio-Temporal Flip Augmentation to reduce bias.
- The experiments and analyses are relatively comprehensive.

**Weaknesses:**

- Video sources are limited; it is unclear whether the dataset is sufficiently diverse. A thorough analysis of scene diversity is needed.
- The question templates are not clearly enumerated or shown; this is important. An analysis of question diversity is needed.
- The manual verification process is insufficiently described; clarify procedures and justify data quality guarantees.
- Evaluating dedicated spatial understanding models would strengthen the work, though it is not strictly necessary.
- In Figure 1, the last category appears mislabeled.
- Text in tables is too large while text in figures is too small.

**Questions:**

- Dataset diversity: Provide source distributions and scene/category stats; add a diversity analysis.
- Question templates: List all templates with examples and counts; analyze template and question diversity.
- Quality controls: Describe the quality-control process in detail. It would also help to report human performance on DSI-Bench.
- Model coverage: Consider evaluating more spatial expertise models, such as [1] [2] [3].
- Fix the mislabeled last category in Figure 1 (if mislabeled). Reduce table font size and enlarge figure text for readability.

[1] Wu, Diankun, et al. "Spatial-mllm: Boosting mllm capabilities in visual-based spatial intelligence." arXiv preprint arXiv:2505.23747 (2025).

[2] Deng, Nianchen, et al. "InternSpatial: A Comprehensive Dataset for Spatial Reasoning in Vision-Language Models." arXiv preprint arXiv:2506.18385 (2025).

[3] Fan, Zhiwen, et al. "VLM-3R: Vision-Language Models Augmented with Instruction-Aligned 3D Reconstruction." arXiv preprint arXiv:2505.20279 (2025).

---

### Official Review · Reviewer_dqES · 2025-11-01

**Soundness:** 3
**Presentation:** 3
**Contribution:** 3
**Rating:** 6
**Confidence:** 4

**Summary:**

This work introduces a benchmark that focuses on perspective taking in images/video understanding.  Both the observer (photographer) and the agent's perspective are queried. Data is categorized based on task/observer/object properties/changes and augmented (e.g temporal reversal, etc). An appropriately comprehensive and high-performing baselines are compared.

**Strengths:**

The problem is natural, necessary, and clearly not one that current models can handle. The examples are also nicely explanatory of the task.

**Weaknesses:**

1. See questions below
2. The model prompts are not presented in the appendix.  I would like a better understanding of how the way data/questions are presented to the models affects performance.  This includes few-shot settings. I will propose an extreme case.  Imagine I gave the the model three of four augmentations in the context.  Would it still fail to predict the fourth on inference? What about just one example?
3. *Minor* -- typos, "genrated" and Fig 1 has ~no caption, backward quotes, etc

**Questions:**

- Am I correct that only multi-choice questions (data annotation pipeline) were used, but Figure 1 includes richer captions?
- Can you expand on the analysis to explain trends in the performance? are the primary errors flips (e.g. left-right) or are there insights into why models perform as they do? [I appreciate the examples, but curious about statistics]
- Provide examples of claim on line 387
- Please expand on the annotation and data validation process

---

### Official Review · Reviewer_PNEE · 2025-11-01

**Soundness:** 3
**Presentation:** 3
**Contribution:** 2
**Rating:** 4
**Confidence:** 3

**Summary:**

The paper introduces DSI-Bench, a video QA benchmark designed to evaluate dynamic spatial intelligence through joint reasoning about self-motion (observer) and object motion in 3D scenes. The benchmark contains 943 short video clips standardized to 480p and over 1,700 manually annotated and verified Q&A pairs spanning six question types across three general task categories (Object–Scene, Observer–Scene, and Observer–Object). Each video is augmented via horizontal flipping and time reversal, yielding four symmetry variants to probe bias and robustness. Evaluation results are reported both sample-wise (treating each variant independently) and group-wise (requiring consistency across variants). The authors evaluate 14 VLMs and two 3D "expertise" models (VGGT, SpatialTrackerV2) under direct answering and free-form reasoning settings. The authors illustrate several key findings through their empirical study, including: (1) dynamic cases are consistently harder than static ones (as expected), (2) free-form reasoning yields marginal and unstable gains, (3) models show a pronounced "forward" selection bias, and (4) VLMs often conflate rotation with translation and entangle object and observer motion.

**Strengths:**

1. Clear Motivation and Well-Defined Problem Setting
The paper is clearly motivated by an under-explored but important challenge: joint reasoning about observer and object motion in dynamic 3D scenes. While prior benchmarks focus mainly on static or single-motion settings, DSI-Bench explicitly targets the coupled dynamics of self-motion and object motion.

2. High-Quality Dataset and Comprehensive Evaluation
The dataset construction process is systematic and rigorous: nearly 1,000 videos are standardized to a consistent format (480p, ~3 s clips), with motion types carefully annotated and verified by human experts. Each video is augmented via spatial-temporal symmetry (flip + reversal) to mitigate bias, and all question-answer pairs are template-based but human-refined to ensure clarity and correctness.
Evaluation is thorough and balanced, covering 14 proprietary and open-source VLMs as well as two 3D expertise models, under both direct answering and free-form reasoning protocols. The shared evaluation setup enables fair and controlled comparisons across diverse model architectures.

3. Insightful and Interpretable Error Analysis
The paper provides clear, model-agnostic analyses of where and why current models fail, identifying several characteristic failure modes, including forward motion bias, rotation-translation confusion, and undecoupled motion reasoning. The paper further provide qualitative visualizations that make the findings interpretable.

**Weaknesses:**

1. Limited novelty comparison in task setup: While the paper’s focus on joint observer–object motion reasoning is distinctive within general VLM benchmarks, similar multi-motion setups have been extensively studied in other domains, such as autonomous driving and embodied simulation (e.g., ego-vehicle vs. surrounding vehicle motion reasoning)[1,2]. The paper would benefit from a clearer comparison and positioning against these prior datasets and tasks, highlighting what makes DSI-Bench fundamentally different.

2. Narrow question diversity and template-based limitations: The current QA design focus almost entirely on motion direction recognition, which, while well-defined, captures only a small slice of dynamic spatial reasoning. Broader capabilities (such as intent or trajectory prediction, relational grounding, multi-object interactions, and counting under motion) are not yet assessed. Additionally, the short 3-second clips constrain long-horizon reasoning about continuous ego-motion or object trajectories. Moreover, because all QAs are template-generated and manually refined, there is a risk of linguistic regularity or pattern bias that models might exploit.

3. Incomplete dataset statistics and analysis: The dataset aggregates clips from multiple heterogeneous sources (CameraBench, Kinetics-700, SynFMC, LLaVA-178K, and web videos), but the paper does not report per-source statistics or performance. A breakdown by source type (synthetic vs. real), motion domain (human, vehicle, object), or environment (indoor/outdoor) would strengthen interpretability and clarify bias sources. Similarly, reporting answer option distributions and motion direction frequencies would help assess whether class imbalance or answer priors influence model results.

[1] Ana-Maria Marcu, Long Chen, Jan Hünermann, Alice Karnsund, Benoit Hanotte, Prajwal Chidananda, Saurabh Nair, Vĳay Badrinarayanan, Alex Kendall, Jamie Shotton, Elahe Arani, Oleg Sinavski. "LingoQA: Visual Question Answering for Autonomous Driving." (2024).

[2] Chirag Parikh, Deepti Rawat, Rakshitha R. T., Tathagata Ghosh, & Ravi Kiran Sarvadevabhatla. (2025). RoadSocial: A Diverse VideoQA Dataset and Benchmark for Road Event Understanding from Social Video Narratives.

**Questions:**

1. The abstract claims “nine decoupled motion patterns”, while Sec. 3.2 says the benchmark covers “5 decoupled motion patterns”, and elsewhere you categorize motion into three types (translation, rotation, combination). Please reconcile these counts and define the motion taxonomy precisely.

2. The group-wise random baseline is reported as 0.05%, but given 4 variants and 4-way choices, the theoretical probability of ≥ 3 correct should be around 5%. Could the authors confirm whether this is a typo or computational error, and clarify its effect on robustness conclusions?

3. Spatial reasoning accuracy often depends heavily on temporal sampling. Why was 5 fps chosen as the standard for all models? Have the authors tested whether fewer frames (e.g., 1 fps) or higher frame rates alter model performance or bias? A short analysis or justification would clarify this design decision.

---

### Official Review · Reviewer_8P7A · 2025-11-02

**Soundness:** 2
**Presentation:** 2
**Contribution:** 3
**Rating:** 2
**Confidence:** 4

**Summary:**

This paper introduced a new spatial intelligence benchmark, DSI-Bench, a benchmark specifically designed for the systematic evaluation of dynamic spatial reasoning. The authors generated complementary samples through spatial–temporal symmetry-based augmentation, to reduce data bias and to analyze model bias and hallucination. Several general-purpose visual-language models and expert visual foundation models are evaluated on DSI-Bench, and error analysis on the failure cases are provided.

**Strengths:**

The DSI-bench is a good effort to benchmark a model's spatial intelligence, especially in more dynamic scenes.

Most existing evaluations focus on static scenes or observers, and the DSI-bench provides a good benchmark for the dynamic scenes.

The spatial and temporal augmentation to reduce the bias is a good effort. Correspondingly, the group evaluation is a good indicator of robustness.

It seems that all annotations are reviewed and confirmed by human annotators, including the augmented ones. This indicates that the data quality may be high. However, the authors may want to provide more details to prove/showcase the data quality.

**Weaknesses:**

1) There paper does not mention existing benchmarks that evaluates dynamic spatial reasoning. While additional benchmarks in this domain are useful, it would be great to compare and contrast the proposed one to the existing benchmarks. For example:

- SAT: Dynamic Spatial Aptitude Training for Multimodal Language Models, COLM 2025

2) The question types are very limited, only a few fixed templates. Why not expand to more questions?

3) The task’s difficulty is not thoroughly studied. The zero-shot performance of popular VLMs and expert models are reported. However, few-shot performance is not studied. For example, the “forward” bias is likely a fine-tuning problem (perhaps with a few examples, the issue can be resolved). The authors may want to study whether few-shot fine-tuning will mostly solve this problem. Or is this problem fundamentally challenging? If so, what is the challenging part?

4) Key details are missing, so cannot concretely evaluate the quality of the benchmark:

The QAs in this benchmark are generated from templates. Each task in Figure 2 should have its own template. It is important to provide all the templates for all the tasks (at least in Appendix).

The question types are very limited (since it comes from templates). The underlying info behind these QAs are from the Annotation block in Figure 3. From Section 3.2, the annotations seem to be discrete (details are lacking there).  The authors should provide details of the annotation tasks in Appendix.

The original response from VLMs (Figure 6&7&8) are missing. The provided responses seem to be the authors’ summary of the VLM’s response. It would be good to provide the original repose in the Appendix so that the readers get a sense of how these models are doing.

Additionally, it is unclear how the authors input videos to GPT5.

The details of “3D Expertise Model Evaluation” are missing. The authors may want to provide enough details so that others can reproduce the results in the paper.

In Figure 6, Gemini-2.5-Pro was used twice. This should be an error.

**Questions:**

1) How does this benchmark compare to existing ones, including SAT?

2) See questions above about the missing details

3) If we fine-tune an expert model on this annotation task with few examples, to what degree will the task be solved? In other words, besides zero-shot evaluation of existing models, some fine-tuning evaluation may be useful to evaluate the difficulty of this task.

**Details Of Ethics Concerns:**

The dataset contains many movie clips, which may be subject to copyright. Authors should demonstrate that they have legal permission to for model evaluation and re-distribution of the datasets, and what if any restrictions exist.

---

### Note · Authors · 2025-11-12

I have read and agree with the venue's withdrawal policy on behalf of myself and my co-authors.